# Studying phonon coherence with a quantum sensor

Agnetta Y. Cleland ®[1], E. Alex Wollack ®[1] & Amir H. Safavi-Naeini ®[1]✉

Nanomechanical oscillators offer numerous advantages for quantum technologies. Their integration with superconducting qubits shows promise for hardware-efficient quantum error-correction protocols involving superpositions of mechanical coherent states. Limitations of this approach include mechanical decoherence processes, particularly two-level system (TLS) defects, which have been widely studied using classical fields and detectors. In this manuscript, we use a superconducting qubit as a quantum sensor to perform phonon number-resolved measurements on a piezoelectrically coupled phononic crystal cavity. This enables a high-resolution study of mechanical dissipation and dephasing in coherent states of variable size ($\bar{n} \simeq 1 - 10$ phonons). We observe nonexponential relaxation and state size-dependent reduction of the dephasing rate, which we attribute to TLS. Using a numerical model, we reproduce the dissipation signatures (and to a lesser extent, the dephasing signatures) via emission into a small ensemble ($N = 5$) of rapidly dephasing TLS. Our findings comprise a detailed examination of TLS-induced phonon decoherence in the quantum regime.

In the field of quantum technology, nanomechanical oscillators offer a host of useful properties given their compact size, long lifetimes, and ability to detect force and motion. These devices hold the potential to serve as long-lived memories for computation[1,2], transducers for communication[3–5], and high-precision quantum sensors[6]. Their ability to interact with superconducting qubits through the piezoelectric effect has allowed mechanical systems to be brought into the quantum regime[7–16]. In order to fully realize the potential of this hybrid platform, it is crucial to understand decoherence processes affecting mechanical oscillators in the quantum limit. For mechanical states with large numbers of phonons ($\bar{n} \gg 1$), the established work has followed semiclassical spectroscopic methods to detect time-averaged energy loss[17–19] and frequency noise[20,21] in the resonator. More recent studies of mechanical devices in the few-phonon regime have used a superconducting qubit to perform single-phonon characterization[8,10,16,22].

One important loss channel affecting nanomechanical systems is two-level system (TLS) defects, similar to those which exist in amorphous glasses and imperfect crystalline materials[23]. TLS can couple to electromagnetic and elastic fields, creating a dissipation channel for nearby modes[24–26]. The dynamics of TLS-induced decay and decoherence at microwave frequencies have been the subject of extensive study in both electromagnetic[20,27–32] and nanomechanical systems[17–19,33–35]. Their contribution to microwave loss has been shown to depend on power and temperature, allowing the TLS-induced loss tangent to be extracted from measurements of the resonator's frequency and quality factor[19,27]. In particular, TLS contribution to dielectric loss is known to be saturable−it is suppressed when the intra-cavity energy exceeds a critical threshold. In this regime, the TLS spontaneous excitation and emission rates equilibrate as the TLS reach a steady state in which they no longer absorb energy from the resonator mode.

Many prior studies of TLS saturation have involved monitoring the scattered response of a resonator for a range of drive powers[19,27,30–32]. Superconducting qubits, with their accompanying suite of readout and control techniques, offer an even more sensitive probe of TLS behavior. This platform has enabled spectroscopy of a TLS ensemble's strain-dependent spectrum[36–38], real-time tracking of their shifting frequencies[39], and even preparation of a single TLS as a quantum memory, with direct measurement of its relaxation and coherence times[40].

[1]Department of Applied Physics and Ginzton Laboratory, Stanford University 348 Via Pueblo Mall, Stanford, CA 94305, USA. ✉e-mail: safavi@stanford.edu

In this work, we aim to extend the flexibility of circuit QED measurement techniques into the domain of mechanical devices to learn about TLS behavior. In our approach, the qubit nonlinearity allows for quantum nondemolition measurement of the mechanical state. We use a Ramsey protocol with a superconducting qubit that allows us to perform time- and phonon number-resolved measurements on the mechanical resonator[22,41]. We leverage this approach to observe signatures of TLS-induced dissipation in the time domain. This technique, which could easily be extended to other bosonic systems, affords us a highly granular picture of how larger mechanical states of an oscillator evolve in time, shedding new light on the quantum nonlinear dissipation processes.

The mechanical dissipation dynamics we observe are consistent with emission into a bath of rapidly dephasing TLS. In ringdown measurements of mechanical coherent states of variable size, we observe an initial period of fast decay which eventually gives way to significantly slower dissipation. Thus, the mechanical dissipation rate changes with the evolving phonon state size, and cannot be described by the simple linear relation expected for a harmonic oscillator. We reproduce this behavior using a simple numerical model incorporating a small ensemble of TLS, where the initial fast decay shows the mechanical mode decaying into the TLS, and slower decay emerges after the TLS become saturated. We also perform mechanical dephasing measurements using an interferometry protocol involving coherent displacements of the mechanical state, which we also study numerically.

## Results and discussion
### Device description

The mechanical oscillators under study are one-dimensional phononic crystal cavities made of thin-film lithium niobate (LN). While the device contains two resonators, these experiments focus on only one cavity, which supports the higher frequency mode (Fig 1a). The cavity is formed from periodic structures, acting as acoustic mirrors, which suspend a defect site from both ends[42]. The periodicity of the mirrors gives rise to a full phononic bandgap which protects the localized mode from phonon radiation channels (Supplementary Fig. 1 and Supplementary Note 1). In addition to the large coupling rates arising from the strong piezoelectric effect of LN, this design approach offers a compelling platform for studies in quantum acoustics, because it robustly removes the effects of clamping and other linear scattering losses. Furthermore, the small mode volume of these cavities make them an ideal candidate for studies of TLS as mechanical loss channels, since it yields stronger coupling to fewer TLS.

We assemble our device in a hybrid flip-chip architecture. The mechanical resonators are fabricated on the top chip, and the bottom chip includes a superconducting qubit, control lines, and a coplanar waveguide readout resonator (Fig. 1b). The qubit is a frequency-tunable planar transmon, with an on-chip flux line providing capability for both static and pulsed frequency control (Fig 1b, inset). The qubit fabrication is described in prior work[11,16] and follows methods developed in ref. [43]. As illustrated in Fig. 1c, the mechanics fabrication process begins with thin-film LN on a silicon substrate. We thermally anneal the sample for 8 hours at 500 C before thinning the LN by argon ion milling to a target final thickness of 250 nm. The nanomechanical structures are then patterned by electron-beam lithography and argon ion milling using a hydrogen silsesquioxane (HSQ) mask. We remove redeposited material from the LN sidewalls in a heated bath of dilute hydrofluoric acid, followed by a piranha etch. Next, we define metallic layers with a combination of electron-beam and photolithography, including bandages to create the relevant galvanic connections. We perform an oxygen plasma descum before each deposition to minimize the presence of polymer residues at metallic interfaces. Finally, we undercut the nanomechanical structures from the substrate with a masked xenon difluoride (XeF$_2$) dry etch. This concludes the mechanical device fabrication, before the LN chip is aligned and bonded to an accompanying qubit chip (Fig. 1d) in a

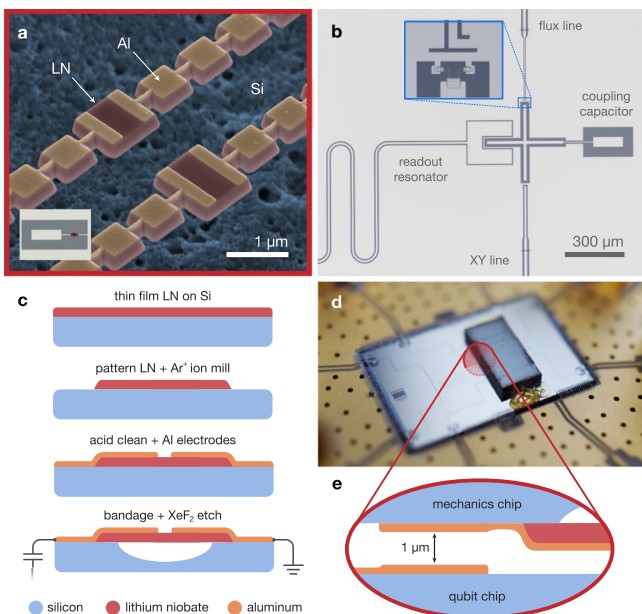

**Fig. 1 | Device and fabrication. a** False-color scanning electron micrograph of the mechanical resonators. Lithium niobate (LN) phononic crystal cavities (red) are patterned with aluminum electrodes (orange) and undercut from the silicon substrate (blue). Inset shows an optical micrograph of the coupling capacitor pad, which is galvanically connected to the mechanical electrodes. The resonators are too small to be seen in this image, but their location is indicated in red. **b** Optical micrograph of the bottom chip, showing the superconducting qubit, control lines, and readout resonator. Inset shows the qubit's SQUID and flux line. **c** Mechanical device fabrication. Cross-sectional illustration of the patterning, cleaning, metallization, and release etching steps. Bandages create electrical contact between the active mechanics electrode and the coupling capacitor pad, as well as between the ground electrode and the chip's ground plane. **d** Photograph of the composite flip-chip assembly. Control lines on the qubit (bottom) chip are visible, as well as the polymer adhesive which fixes the top chip in place. **e** Approximate location of the cross-chip coupling capacitor. The illustration shows the two halves of the coupling capacitor (orange) aligned vertically and separated by approximately 1 $\mu$m.

submicron die bonder[16,44]. The qubit is capacitively coupled to the mechanical mode across a small vacuum gap, which is defined by the flip-chip separation distance (Fig. 1e).

The completed flip-chip package is cooled to a temperature of 10 mK in a dilution refrigerator. As we bias the qubit away from its maximum frequency $\omega_{ge}^{max}/2\pi = 2.443$ GHz, we measure the qubit $T_1$ over a wide tuning range to find a mean value $T_1 = 4.9 \pm 2.3\,\mu$s, with $T_2 = 1.4\,\mu$s at the flux sweet spot. When the qubit is rapidly tuned into resonance with the mechanical mode at $\omega_m/2\pi = 2.339$ GHz, the resulting Rabi oscillations allow us to extract the coupling rate $g/2\pi = 10.5 \pm 0.1$ MHz. Further details on basic device characterization can be found in a prior manuscript[16].

### Phonon number measurements

In the following experiments, we statically bias the qubit near the mechanical mode and use it as a probe to study dissipation and dephasing of mechanical coherent states. We extract information about the mechanical state through the qubit spectrum by means of their dispersive coupling. The qubit and mechanics are coupled through the piezoelectric effect to give an interaction Hamiltonian $\hat{H}_{int} = g(\hat{b} + \hat{b}^\dagger)\hat{\sigma}_x$, with $\hat{b}$ corresponding to the mechanics and Pauli operators $\hat{\sigma}$ corresponding to the qubit. When the two modes are far detuned, the effective Hamiltonian for the system is given by[45]

$$\hat{H}_{eff} = \omega_m \hat{b}^\dagger \hat{b} + \frac{1}{2}\left(\omega_{ge} + 2\chi \hat{b}^\dagger \hat{b}\right)\hat{\sigma}_z. \tag{1}$$

In this limit, the qubit spectrum acquires a frequency shift $2\chi$ for each populated energy level in the mechanical oscillator. The magnitude of this shift depends on the qubit anharmonicity $\alpha_q$, mechanical coupling rate $g$, and detuning $\Delta = \omega_{ge} - \omega_m$ as

$$\chi = -\frac{g^2}{\Delta}\frac{\alpha_q}{\Delta - \alpha_q} . \quad (2)$$

Since this value varies with the qubit frequency, it is important to choose an operating point which yields a large $\chi$ while keeping the system in the strong dispersive limit. In this regime, $\chi$ exceeds all decoherence rates of the system, but the detuning is sufficiently large to prevent significant linear coupling between the two modes. For our experiments, we statically bias the qubit to $\omega_{ge}/2\pi = 2.262$ GHz, which corresponds to $|\Delta| \approx 8g$ and a measured dispersive shift $2\chi/2\pi = -1.48 \pm 0.05$ MHz.

When the mechanical mode is populated, we can perform a Ramsey measurement on the qubit to resolve the array of dispersive shifts and thus obtain the full phonon number distribution of the mechanical state. The Ramsey signal $S(t)$ takes the form of a sum of oscillating terms whose frequencies differ by integer multiples of $2\chi$, indexed by the Fock number $n$, with an exponentially decaying envelope:

$$S(t) = \sum_{n=0}^{n_{max}} A_n e^{-\kappa t} \cos\left((\omega_0 + 2\chi n)t + \varphi_n\right) . \quad (3)$$

We fit the Ramsey signal to Eq. (3) with fit parameters $A_n$, $\kappa$, and $\chi$. We extract the phonon number distribution from the normalized amplitudes, $P(n) = A_n/\sum_n A_n$. In contrast to previous work[16] where the exponential decay rate in Eq. (3) was taken to depend linearly on $n$, we choose here to model decay as independent of the phonon occupation number (Supplementary Note 2).

## Phonon-resolved decay

We use the Ramsey protocol to study decay of mechanical coherent states. We first apply a displacement pulse $\hat{D}_\alpha$ at the mechanical frequency to the qubit's XY line (Fig. 2a), which drives the mechanical mode into a coherent state. After a variable delay $\tau$, we perform a Ramsey measurement on the qubit to extract the mechanical $P(n)$, as described above (Fig. 2b,c). From this, we calculate the average number of phonons, $\bar{n} = \sum_{n=0}^\infty n \cdot P(n)$. Repeating this sequence for a range of $\tau$ values constitutes a *phonon-number resolved* ringdown measurement of a mechanical coherent state (Fig. 2d). The measurement provides a high degree of resolution, revealing not only the decaying mean phonon number in the resonator, but also how the distribution evolves in time (Fig. 2e and Supplementary Fig. 2).

We observe that the mechanical dissipation follows a double-exponential trajectory, with an initial fast decay followed by significantly slower relaxation. Similar multi-exponential behavior appears in single-phonon measurements in a prior study of the same device[16]. By varying the duration and amplitude of the initial displacement pulse, we study how the energy decay and saturation dynamics depend on the size of the initial mechanical state (Fig. 2d). We fit each decay curve to a double-exponential form:

$$\bar{n}(\tau) = a_1 e^{-\kappa_1 \tau} + a_2 e^{-\kappa_2 \tau}. \quad (4)$$

We examine the behavior of the fast decay $\kappa_1/2\pi \simeq 30-70$ kHz, slow decay $\kappa_2/2\pi \simeq 1-3$ kHz, and their corresponding weights $a_1$, $a_2$ as we vary the initial mean phonon number $\bar{n}_0 = \bar{n}(\tau = 0)$.

Nonexponential decay has been observed in graphene mechanical resonators operated in a strongly nonlinear regime[46]. It has also been known to occur in superconducting qubits due to fluctuations in quasiparticle population[47] and, more recently, due to interactions with

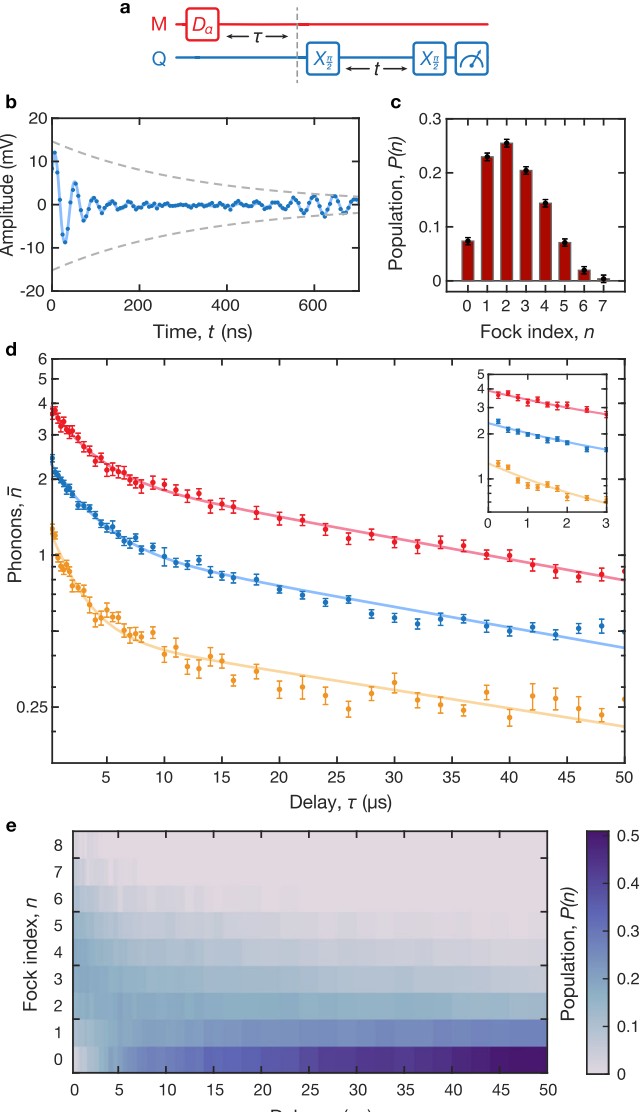

**Fig. 2 | Energy decay measurement. a** Pulse sequence showing a displacement $\hat{D}_\alpha$ which prepares a mechanical coherent state (left) and mechanical state readout using a Ramsey measurement (right) after a variable delay $\tau$. The Ramsey measurement consists of two $\pi/2$ rotations of the qubit state, separated by a variable delay $t$. **b** Representative time domain trace from a Ramsey measurement. The data (points) and a fit to Eq. (3) (solid line) are shown, along with the fitted exponential decay envelope (dashed line). **c** Phonon number distribution extracted from the time domain data in **b**. This is used to calculate the mean phonon number, $\bar{n}$. All error bars represent one standard deviation. **d** Sample ringdown measurements $\bar{n}(\tau)$ with varied initial state sizes, $\bar{n}_0 = \bar{n}(0)$. Inset shows an expanded view of the early-time data, $\tau \le 3\,\mu s$. **e** Evolution of phonon number distribution for an initial state size $\bar{n}_0 = 3.89$ (red data set in **d**). The color plot indicates the population in each Fock level at each point in time.

TLS[48,49]. Neither quasiparticles nor mechanical nonlinearity can plausibly lead to the nonexponential decay observed in our experiments. We argue that the multi-exponential energy decay in our system is caused by resonant interaction of the mechanics with a handful of rapidly dephasing TLS. The initial fast decay occurs as the mechanical mode emits energy into the TLS, and we note that we do not observe any coherent oscillations between the two coupled systems. As the TLS become populated, they lose their ability to absorb more phonons, thus becoming saturated. Consequently, the rapid decay ends and slower decay emerges, caused by either energy relaxation of the TLS or other dissipation processes affecting the resonator. We perform

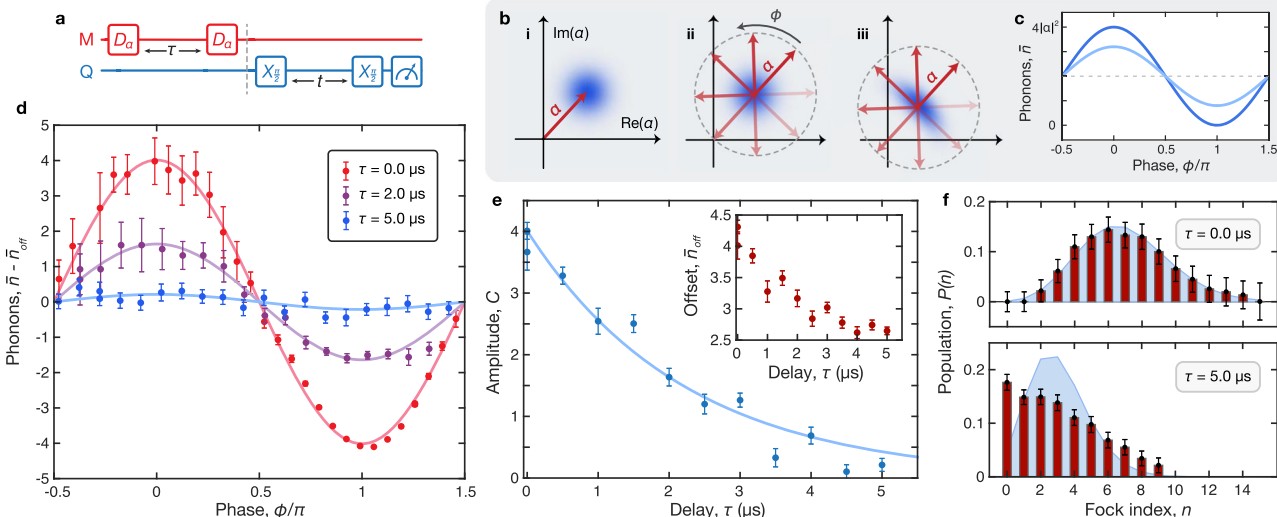

**Fig. 3 | Displacement interferometry. a** Pulse sequence for the dephasing measurement showing interferometry protocol (left) and mechanical state readout (right). We apply two displacement pulses $\hat{D}_\alpha$ separated by a variable delay $\tau$ and with a programmed relative phase $\phi$. Last, we perform a Ramsey measurement to characterize the resulting mechanical state for each value of $\phi$ and $\tau$. **b** Illustration of the measurement principle. **i** The initial displacement $\hat{D}_\alpha$ excites a coherent state. **ii** The resulting states after the second displacement trace a circular path in phase space. In the absence of dephasing ($\tau \ll T_{2m}$), this path intersects the origin when $\phi = \pi$. **iii** In the presence of dephasing ($\tau \gtrsim T_{2m}$), this circular path is shifted in phase space. **c** Illustrated measurement result, showing pure dephasing in the absence of relaxation. For each value of $\tau$, the average phonon number $\bar{n}$ depends sinusoidally on $\phi$. The case with no dephasing (**b**, **ii**) is plotted in dark blue, while the dephased case (**b**, **iii**) is shown in light blue. **d** Experimental results. We plot the

extracted phonon number (points) and corresponding fit to Eq. (5) (line) for a few values of $\tau$ with a common initial state size. **e** Sample dephasing ringdown measurement. We plot the decaying amplitudes (points) extracted from the interference fringes and fit them to an exponential decay function (line) to extract the mechanical dephasing rate $\gamma_{2m} = 1/T_{2m}$. **f** Dephasing effects in phonon distribution. Red bars indicate the Fock level occupations extracted from Ramsey data. Each panel corresponds to one data point $\bar{n}$ in (**d**). The shaded blue curve shows the $P(n)$ distribution of the coherent state with the same mean phonon number. The top panel ($\tau = 0$ and $\bar{n} = 6.97$) shows a mechanical state which closely follows a coherent state distribution, while the state in the bottom panel ($\tau = 5.0\,\mu s$ and $\bar{n} = 3.06$) diverges notably from this distribution due to mechanical dephasing. All error bars represent one standard deviation.

numerical modeling to reproduce the salient features of this process, described later in this paper.

## Phonon-resolved dephasing

A modification of our measurement protocol allows us to extract the mechanical dephasing time $T_{2m}$ using coherent states of motion with varying amplitude. As shown in Fig. 3a, we now displace the mechanical state twice: once to initialize the measurement with a coherent state, and again after a variable delay $\tau$. The two displacement pulses have identical amplitude and duration, but the second pulse has a programmed phase $\phi$ that is swept from 0 to $2\pi$ for each value of $\tau$. The protocol concludes with a Ramsey measurement to characterize the final mechanical state.

We illustrate the principle behind this displacement interferometry protocol in Fig. 3b. The first displacement initializes a coherent state (Fig. 3b,i). After the second displacement, the resulting mechanical states lie on a circle in the complex plane. In the absence of dephasing ($\tau \ll T_{2m}$), maximal displacement $2\alpha$ occurs at $\phi = 0$, and the final state is returned to the origin when the pulses are out of phase by $\phi = \pi$ (Fig. 3b, ii). In the presence of dephasing ($\tau \gtrsim T_{2m}$), this resulting path is shifted in the complex plane and no longer intersects the origin (Fig. 3b, iii). In these measurements, the final state size $\bar{n}$ depends sinusoidally on $\phi$, and the effect of dephasing can be seen in the reduced oscillation amplitude for larger delays (Fig. 3c). These amplitudes decay exponentially in time at a rate $\gamma_{2m} = 1/T_{2m}$ (Supplementary Note 3). We fit the interference data to the following form to extract the amplitude $C$ and offset $\bar{n}_{off}$:

$$\bar{n}(\phi) = C \cos(\phi + \phi_0) + \bar{n}_{off}. \tag{5}$$

The phase offset $\phi_0$ is dependent on the delay $\tau$, and could arise due to frequency uncertainty.

A few representative interferometry traces are shown in Fig. 3d for an initial state $\bar{n}_0 = 2.29$. Each interferogram is plotted relative to its offset $\bar{n}_{off}$ for visual clarity. We note that the offsets also decay in time at a rate $\gamma_{1m} = 1/T_{1m}$, but the timescale of the interferometry data is insufficient to accurately determine this value. In Fig. 3e we plot the extracted offsets $\bar{n}_{off}$ (inset), amplitudes $C$ and exponential fit for this data set, from which we extract $T_{2m} = 2.2 \pm 0.2\,\mu s$.

The effect of dephasing is also visible in the phonon number distribution of the mechanical oscillator. In principle, a coherent state undergoing amplitude decay should remain a coherent state and therefore maintain a Poisson distribution. However, in the presence of dephasing, the state evolves into something other than a coherent state whose $P(n)$ may deviate significantly from the Poissonian form. This effect is evident in Fig. 3f, where we examine $P(n)$ extracted from data for two representative states: one with $\tau = 0$ (top) and another with $\tau = 5.0\,\mu s$ (bottom). Each distribution is compared to a coherent state $|\beta\rangle$ whose $P(n)$ is given by a Poisson distribution with the same average phonon number as the data, $|\beta|^2 = \bar{n}$. The top panel shows good agreement between the data (red bars) and the coherent state distribution (shaded blue), while the bottom panel shows a pronounced divergence between the two. We observe a similar effect in a simulation of the dephasing process (Fig. 4f).

We repeat the dephasing measurement for three distinct initial states and find, perhaps surprisingly, that the dephasing rate is reduced for larger phonon states (Fig. 4e). In a naive model, we expect the mechanical frequency jitter to be either primarily due to fluctuating off-resonant TLS, which would not be saturated by weakly driving the mechanics, or from excitation and relaxation of resonant TLS, which occur even when they are saturated. Saturation implies that the rates of phonon emission and absorption into the TLS bath are roughly equal, canceling their effect on energy decay and leading to longer $T_{1m}$ as we observe. However, dephasing should still occur if emission and

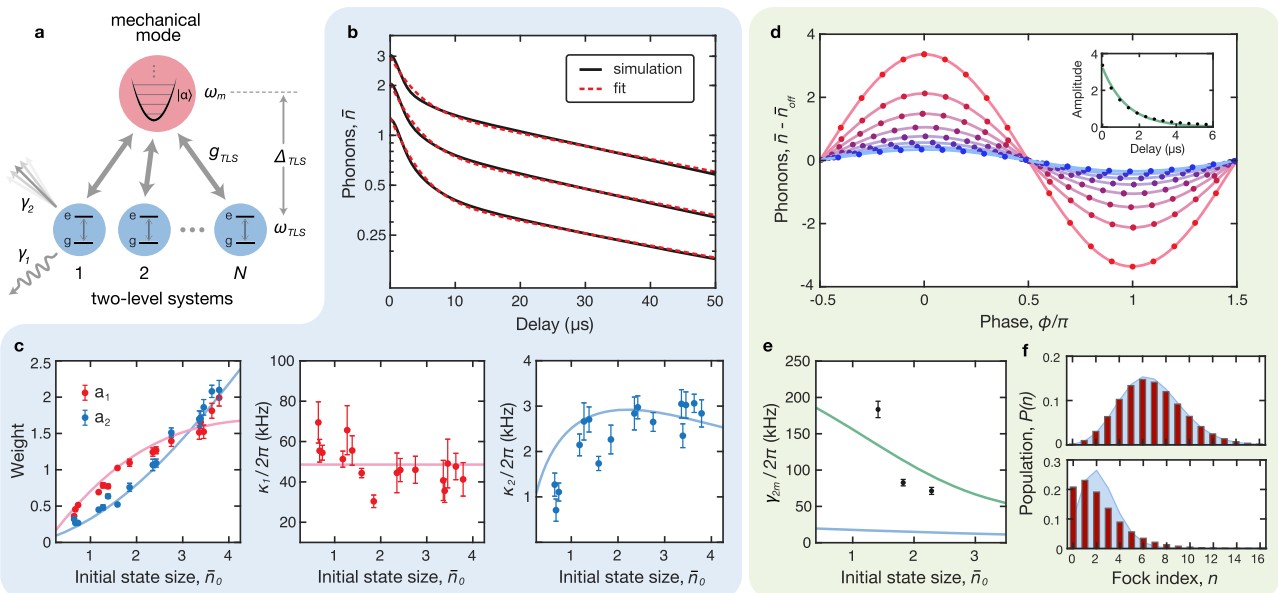

**Fig. 4 | Modeling dissipation and decoherence. a** Schematic of the simulation showing a collection of $N$ two-level oscillators coupled at a rate $g_{TLS}$ to the mechanical mode. The TLS are detuned from the mechanical mode by $\Delta_{TLS}$ and have an intrinsic decay rate $\gamma_1$ and dephasing rate $\gamma_2$. **b** Simulated mechanical state evolution showing TLS-induced dissipation for a range of initial state sizes. The mechanical energy decay, given by the average phonon number $\bar{n}$ (black), follows a double-exponential trajectory, which we fit to Eq. (4) (red). **c** Energy decay fit metrics from experimental data (points) and simulation (line) corresponding to the fast initial decay (red) and slow secondary decay (blue) are shown for a range of initial state sizes $\bar{n}_0$. The left panel shows the exponential weights $a_1$ (red) and $a_2$ (blue). The middle panel shows the experimentally observed fast decay rates $\kappa_1$ (points) and their mean value (line). The right panel shows the fit results from experiment (points) and simulation (line) for the slow decay rate $\kappa_2$. All error bars represent one standard deviation. **d** Representative result of the interferometry protocol for displacement amplitude $\alpha = 1.30$. The simulated phonon states (points) are fit to Eq. (5) (lines) for a range of delays in the time-evolved mechanical state. Inset shows the extracted amplitudes (points) and fit to exponential decay (line) from which we extract $\gamma_{2m}$. **e** The experimentally observed dephasing rates (points) are compared to simulation results (lines) computed using two different coupling strengths $g_{TLS}$. The weak coupling limit ($g_{TLS}/2\pi = 33$ kHz) corresponding to the results in (**b**, **c**) is shown in blue, while a larger coupling rate ($g_{TLS}/2\pi = 0.33$ MHz) corresponding to (**d**, **f**) is shown in green. **f** Effect of dephasing on $P(n)$. The simulated phonon distribution (red) is compared to the corresponding coherent state distribution (blue) at short delays (top, $\tau = 0$) and long delays (bottom, $\tau = 7.0$ µs).

absorption of phonons are occurring incoherently. An increase in $T_{2m}$ may therefore point to coherence of the TLS bath with the driving field. We note that a power-dependent reduction of TLS-induced dephasing processes is predicted by the tunneling model proposed by Faoro and Ioffe[50]. The predictions of this model have found good agreement with experimental studies of superconducting resonators which cannot be described by the standard tunneling model.

## Numerical model of mechanics-TLS interaction

To better understand these decay and decoherence signatures, we develop and study numerically[51] a highly simplified model of a mechanical mode interacting with a small number of TLS[52]. Our model includes a collection of $N$ identical two-level oscillators, with annihilation operators $\hat{a}_k$, coupled at a rate $g_{TLS}$ to a harmonic oscillator (Fig. 4a). The TLS have intrinsic decay and dephasing rates $\gamma_1$ and $\gamma_2$, respectively, and are detuned from the mechanical frequency by $\Delta_{TLS}$. We consider only near-resonant TLS, and we assume there is no TLS-TLS coupling. In the frame of the TLS, the system Hamiltonian is given by:

$$\hat{H} = \Delta_{TLS}\hat{b}^\dagger\hat{b} + \sum_{k=1}^{N} g_{TLS}(\hat{a}_k^\dagger\hat{b} + \hat{b}^\dagger\hat{a}_k). \quad (6)$$

We initialize the mechanics and TLS in thermal states, each populated to a level $n_{th} = 0.05$. Next, we prepare the mechanical mode in a coherent state with an instantaneous displacement $\hat{D}_\alpha$ with variable amplitude. We then allow the system to freely evolve under the action of the Hamiltonian in Eq. (6) for a total duration of 50 µs, with collapse operators acting on the TLS. The operators $\sqrt{\gamma_1(n_{th}+1)}\hat{a}_k$ and

$\sqrt{\gamma_1 n_{th}}\hat{a}_k^\dagger$ account for spontaneous emission and absorption in the TLS, while $\sqrt{\gamma_2/2}\hat{a}_k^\dagger\hat{a}_k$ accounts for their dephasing. Dissipation and decoherence of the phonon mode are induced by interactions with the TLS; we do not include collapse operators for the mechanical mode itself.

The numerical solution to this Lindblad master equation returns the density matrix of the total system for every point in time. We examine the evolution of the mechanical mode's $P(n)$ and state size $\bar{n}$. Similar to the experiment, the model shows approximately double-exponential energy decay in Fig. 4b, with weights and rates that depend on the initial state size. We fit each simulated decay profile to Eq. (4); to constrain the fit, we fix $\kappa_1$ to the mean of the experimentally observed values while allowing $a_1$, $a_2$ and $\kappa_2$ to vary.

Although the measurements are insufficient to precisely determine the system parameters $N$, $g_{TLS}$, $\Delta_{TLS}$, $\gamma_1$ and $\gamma_2$, they provide guidance in our choice of values, and the resulting model helps to qualitatively understand the experimental data. It is possible to estimate the microscopic Hamiltonian parameters of the TLS from properties of the host material and the mechanical eigenmode (Supplementary Note 4 and Supplementary Fig. 3). From the expected distribution of these parameters, the model produces reasonable agreement with experiments for a range of values. In Fig. 4b, c we plot the results for $N = 5$, $g_{TLS}/2\pi = 33$ kHz and $\Delta_{TLS}/2\pi = 100$ kHz. The lack of evidence of coherent mechanics-TLS energy exchange suggests that the TLS are rapidly dephasing, $\gamma_2/g_{TLS} \simeq 20 - 30$. For this simulation, we use $\gamma_2/2\pi = 660$ kHz and $\gamma_1/2\pi = 4.0$ kHz. In Fig. 4b we plot the simulated energy decay trajectories (black) and corresponding fits (red) for a few initial states, and Fig. 4c shows good agreement between the fit metrics extracted from these simulations (lines) and the experimental

data (points). See Supplementary Fig. 3 for the corresponding TLS behavior. Moreover, the trend in the weights $a_1$ and $a_2$ suggests that for a sufficiently large initial state size, the contribution of the fast initial decay ($a_1$) saturates while the slow secondary decay ($a_2$) continues to increase. This indicates that above a certain threshold, the physics would eventually be described by single-exponential decay with the slower decay rate $\kappa_2$, as we would intuitively expect once the TLS defects become saturated. This behavior is qualitatively different from the double-exponential decay model that has been used to capture effects due to quasiparticles[47].

To model the dephasing experiment, we apply a second displacement operation with phase $\phi \in [0, 2\pi]$ at a range of delays in the time-evolved mechanical state trajectory. Following the same procedure as before, we extract the sinusoidal amplitudes from the phonon interference fringes (Fig. 4d) and determine the TLS-induced $\gamma_{2m}$ from the decaying amplitudes (Fig. 4d, inset). The results shown in Fig. 4d, f are computed with a stronger coupling rate ($g_{TLS}/2\pi = 0.33$ MHz, $N = 5$, $\gamma_1/2\pi = 4.0$ kHz, $\gamma_2 = 20g_{TLS}$ and $\Delta_{TLS} = 3g_{TLS}$). In Fig. 4e, we compare the extracted $\gamma_{2m}$ for these parameters to the experimental data, as well as to the weakly-coupled simulation results. For the weak-coupling parameters which match the energy decay experiments well, we find that the extracted $\gamma_{2m}$ (blue line) is significantly lower than the experimentally observed values (points). With the stronger coupling rate, $g_{TLS}/2\pi = 0.33$ MHz, the extracted $\gamma_{2m}$ values (green line) fall within the range of our experimental observations. We note that for both values of $g_{TLS}$, the model predicts a reduced dephasing rate for larger phonon states. Finally, in Fig. 4f we show the simulated $P(n)$ for a short delay (top) and longer delay (bottom), where the dephasing effects can be seen as a deviation from the coherent state distribution.

The discrepancy in optimal simulation parameters suggests that our model provides an incomplete picture of the decoherence processes. For instance, it is possible that the mechanical mode suffers from additional dephasing channels, such as thermal excitation of low frequency TLS. It is also possible that more complicated bath dynamics, such as a distribution of non-identical defects with TLS-TLS interactions, are necessary to more accurately describe our data.

In conclusion, we have applied a dispersive phonon number measurement to study coherent states in a phononic crystal resonator. We have examined how the energy decay and dephasing of these states depend on the initial phonon state size, and have reproduced some of these signatures using a simple model incorporating an ensemble of saturable TLS. Future studies would benefit from a more complex computational model, as well as additional measurements to probe the saturation dynamics, such as spectral hole burning[35]. Our results have direct relevance for bosonic error-correction schemes involving coherent states of phonons[2], and highlight the need for improved fabrication and materials processing techniques to mitigate the presence of TLS. Furthermore, this measurement technique builds a foundation for further explorations of fundamental physics in quantum acoustic systems, including observing quantum jumps in mechanical states[53].

## Data availability
The datasets generated and analyzed for the current study are available from the corresponding author upon request.

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

## Acknowledgements

The authors would like to thank M. I. Dykman, M. L. Roukes, O. J. Painter, and P. Winkel for useful discussions. The authors also thank R. G. Gruenke, M. Maksymowych, K. K. S. Multani, N. R. Lee, T. Makihara, and O. Hitchcock for their assistance. We acknowledge support from Amazon Web Services. This work was funded by the U.S. government through the Office of Naval Research (ONR) under grant No. N00014-20-1-2422, the U.S. Department of Energy through Grant No. DE-SC0019174, and the National Science Foundation CAREER award No. ECCS-1941826. A.Y.C. was supported by the Army Research Office through the Quantum Computing Graduate Research Fellowship as well as the Stanford Graduate Fellowship. E.A.W. was supported by the Department of Defense through the National Defense & Engineering Graduate Fellowship. Device fabrication was performed at the Stanford Nano Shared Facilities (SNSF), supported by the National Science Foundation under grant No. ECCS-1542152, and the Stanford Nanofabrication Facility (SNF).

## Author contributions

A.Y.C. and E.A.W. designed and fabricated the device and performed the experiments. A.Y.C. analyzed the data with E.A.W. assisting. A.Y.C. performed the simulations and wrote the manuscript with A.H.S-N. assisting. A.H.S.-N. supervised all efforts.

## Competing interests

A. Y. Cleland is currently a research scientist at Google, E. Alex Wollack is currently a research scientist at Amazon, and A. H. Safavi-Naeini is an Amazon Scholar.
