## [Peer Review File · Nature Communications]

REVIEWER COMMENTS

Reviewer #2 (Remarks to the Author):

The authors present a beautiful new method to study mechanical dissipation and dephasing due to two-level-fluctuators. It relies on a piezo electrically coupled superconducting qubit in the strong dispersive limit - an experimental device platform that has been developed by the same group over the last few years. This has already led to a number of high profile results and I do not see major differences in the device. However, in this work however it is used to analyze mechanical dissipation mechanisms in a novel way.

An improved understanding of TLS and their detrimental effects in coherent quantum circuits is of utmost importance. The presentation is excellent. I would therefore support publication in Nature Communications if the following questions and comments can be adequately addressed.

General questions that should be answered and addressed more clearly in the manuscript:

- 1) What new insights does this study teach us about the microscopic origin of the TLS plaguing both mechanical and electromagnetic (EM) microwave circuits? Dangling bonds vs. localized strain - are the TLS coupling to EM modes and mechanical modes possibly the same?
- 2) To which extent can such a study based on a piezo-electric material distinguish the two, given that any excitation concerns both? Related question: which fraction of the energy of the oscillator energy is actually "mechanical"?
- 3) The authors say in the abstract that previous studies were in the classical domain but there are numerous studies on qubits probing TLS defects either directly, indirectly, i.e. affecting mostly the resonator loss, and in both cases strongly and weakly coupled. In fact, superconducting qubits have become a standard tool to probe and control TLS - in some cases with fairly long coherence. So I think this needs to be put better in context or made more precise. The introduction and reference list is focusses strongly on piezo-electric systems and would also benefit from including some of those studies - both early and recent.

4) Given the importance of the results for the future of bosonic qubit encodings in microscopic mechanical oscillators I am surprised that there is no thorough discussion or outlook.

- What do the measured mechanical energy relaxation and dephasing times tell us?
- Are the results better or worse than the ones in the literature? Can one compare to the equivalent $\tan(\delta)$ of relevant and well studied materials (piezo or not)?
- Why are they better or worse?
- Where are the TLS and are the numbers only relevant for a certain geometry or material?
- What are these timescales limited by and how can they be improved?
- How much do they need to be improved to do something relevant (e.g. those applications mentioned in the conclusion)?
- How realistic is this?

5) The authors mention 2 studies that observed non-exponential energy relaxation of qubits due to other reasons but there are also studies showing this due to TLS. A few recent examples from exponential to bi-exponential to oscillatory (depending on coupling strength to the strongest coupled TLS) are <https://arxiv.org/abs/2204.00499>, <https://arxiv.org/abs/2206.14104>, <https://www.nature.com/articles/ncomms7182>, <https://www.nature.com/articles/ncomms7182>.

With regards to that, can the authors clarify why Eq. 2 (biexponential) is more appropriate compared to a double exponential used in Ref. 12 to describes a primary fast dissipation channel (e.g. the strongest coupled TLS) together with a slower one (e.g. the ensemble formed by the other weaker coupled TLS together with other residual loss)? Is it related to the fact that here they study the relaxation of a harmonic oscillator? Or would both be largely equivalent and equally compatible to the numerical model? Related to that, what is the physical interpretation of the two coefficients a_1 and a_2 in Eq. 2?

In Fig. 4b it looks like that an actual double exponential $n_0 \text{Exp}[a (\text{Exp}[-\kappa_1 t] - 1)] \text{Exp}[-\kappa_2 t]$ (as derived in Ref. 12 for quasiparticle induced decay) rather than a biexponential as in Eq. 2 (maybe consistent naming would help) might fit better with fewer parameters as the fast decay of the numerics appears to be consistently faster (a log scale would be beneficial to better see this). One would obtain different fitted time constants however - would the interesting size dependence observation still hold?

6) It is unclear to me how the randomness and the known long term time variation of the TLS properties play a role in both the experiments and numerical simulations.

- Are these averaged measurements? If yes, how many averages and over what time scales are they obtained? Do the authors see differences when the experiment is repeated?

- Are the results an average of multiple initializations? Are the TLS parameters drawn randomly from a defined probability distribution of realistic detuning, couplings and decoherence times? If yes which distribution? If not, how robust is the observed double exponential behavior? How robust is the dephasing increase with n_0 (or under which assumptions/conditions is this predicted)?

7) One important new aspect is the size dependence of the observed dissipation including the not so intuitive finding that larger states both dephase and decay slower. Even though this is mentioned a few times and even though in the text a size dependent analysis is promised, I do not see clear data, figures, analysis and discussion of this aspect when it is mentioned. Later in Fig. 4c one can see that both weights increase - what does this mean? The primary decay tends to decrease with large error bars (is that meant?) and the secondary decay actually increases with size.

The same for the dephasing, I fail to see the data for the discussion in line 155 ff (I presume Fig.4e is referred to). Is it statistically significant compared to variations, e.g. for various repetitions or averaging times of the experiment? or between devices?

Specific minor comments that should be addressed:

- abstract: is->are

- abstract: I'd mention piezo-electrical coupling

- references are not in order

- introduction: please define the emphasized "quantum nonlinear dissipation" term and how this is applicable here and different to before

- the device description contains quite some fabrication details - are they new and needed at this point?

- Fig. 2, maybe clarify early on that \bar{n} is the mean photon number and consider explicitly indicating n_0 in the plot (or have a legend). Also, how large is the deviation from the Poissonian distribution or is it imposed to be that in the fit? Is that deviation part of the error bars shown in the measured distribution?

- line 90: what would be too much coupling? Also, I'd assume similar measurements and insights could be obtained in the resonant limit?

- line 99: Is the constant κ choice needed to fit the data successfully? Or in other words, was the linear n dependent κ needed to fit earlier (smaller phonon number) Ramsey measurements to successfully fit the data?
- line 108: is there something we learn from the observed time evolution of the distribution? e.g. Poissonian vs. thermal?
- line 112: saturation dynamics vs. size is promised but I miss to see data how the 4 fit parameters in Eq. 2 change vs. initial phonon occupancy as well as (or at least or) a qualitative interpretation of it. also, define "saturation dynamics".
- line 121: The explanation given requires that the T_1 time of the TLS are longer than the energy relaxation time of the mechanical oscillator - is this correct?
- line 134: isn't it mostly energy relaxation that causes the shift of the circle (center) rather than dephasing? Also in Fig. 3biii.
- Is Fig. 3c plotted assuming $1/T_1=0$?
- Fig.3e inset. Does the result qualitatively agree with the expectation of T_1 ?
- line 161: does that mean the TLS bath would be locked to the coherent drive?
- line 177,178: "Similar to the experiment, the model shows an ..."

Reviewer #3 (Remarks to the Author):

In this work the author uses a hybrid micromechanical- SC qubit qubit system to study phonon coherence and the effect of TLS. The qubit is only used as sensor (for Ramsey spectroscopy of the mechanical resonator) so -unless I am missing something- it is not actually very important as such; this is really a paper about TLS in mechanical systems which is a field that in some ways goes back to the 70s with some very impressive experiment having been done in recent years.

This is a clever and in many ways impressive experiment. Getting all the pieces of such a complicated experiment to work together and get reasonable data is far from trivial.

Focus in this work has been on looking on how the energy decay and dephasing depends on the initial phonon state size. While this is an interesting problem the results are -as the authors also state- somewhat inconclusive. The simple model used by the authors does agree with some of the data, but there are only deviations, and I am not entirely convinced by some of the parameters they extract (please see below). This means that while the data looks very nice, I am not entirely sure

what we're meant to learn about TLS physics from the paper. That said, the measurement technique is very interesting.

A few specific comments:

-Double exponential decay is frequently seen in SC qubits, and it is well-known this can be due not only to quasiparticles (which is stated in the manuscripts) but also TLS. Hence, this in itself is not new (although this is a mechanical system, so there is a difference)

-The authors do not really discuss the effects of temperature in the paper. With a fridge temperature of 10 mK we would expect the sample temperature to be somewhat higher (say 50 mK). Now, this corresponds to an energy of about 1 GHz. We would therefore expect all "simple" two-level systems with Δ much lower than this to be saturated (of course we can still have "classical" fluctuators, but they would not behave as simple TLS).

Now, using their model the authors extract values for their 5 strongly coupled TLS of about 100 kHz ($\ll 1$ GHz).

I am quite possibly missing something here, but I don't understand how this works. Again, even at 10 mK we know there are fluctuating TLS (sometimes known as fluctuators), but these are "classical" and not modelled in the way it is done in this paper (they are usually seen because they interact with TLS with much higher energies that are coherent at 10 mK)

Again, maybe I am misunderstanding something, but regardless I don't think this is clearly explained in the paper.

Response to Referees

Nat. Comms. – Studying phonon coherence with a quantum sensor

Black font = referee comments

Blue font = author response

Reviewer #2

The authors present a beautiful new method to study mechanical dissipation and dephasing due to two-level-fluctuators. It relies on a piezo electrically coupled superconducting qubit in the strong dispersive limit - an experimental device platform that has been developed by the same group over the last few years. This has already led to a number of high profile results and I do not see major differences in the device. However, in this work however it is used to analyze mechanical dissipation mechanisms in a novel way.

An improved understanding of TLS and their detrimental effects in coherent quantum circuits is of utmost importance. The presentation is excellent. I would therefore support publication in Nature Communications if the following questions and comments can be adequately addressed.

General questions that should be answered and addressed more clearly in the manuscript:

1) What new insights does this study teach us about the microscopic origin of the TLS plaguing both mechanical and electromagnetic (EM) microwave circuits? Dangling bonds vs. localized strain - are the TLS coupling to EM modes and mechanical modes possibly the same?

Our study provides valuable insights into the nature of TLS defects affecting nanomechanical resonators, though certain microscopic details remain unclear:

- The observed doubly-exponential decay dynamics match well with expectations for phonon emission into a small number of rapidly dephasing TLS. This is supported by separate measurements of the material's loss tangent and fits standard tunneling models of TLS.
- However, the excess dephasing at low phonon numbers and its non-standard saturation with increasing state size hints at physics beyond the standard tunneling model of TLS. There may be additional decoherence processes at play. This effect has been elucidated by our qubit-based measurements.

- The results are consistent with highly localized, strain-sensitive TLS defects that share similarities with those studied in superconducting circuits. But a definitive statement about the microscopic origins requires further materials and surface-sensitive studies.
- In general, TLS can couple differently to electric vs strain fields. Some may couple strongly to strain while being "dark" to electric fields. In a piezoelectric material like LN, the coherent coupling between strain and electric fields further complicates the situation.

In summary, while our measurements provide valuable new insights into TLS-induced decoherence, the microscopic details of the TLS origins and couplings remain an open question requiring further systematic study through measurements and material science. A study of TLS, and a complete understanding of their origins and resulting phenomena is an ongoing effort in the field requiring a multifaceted and multi-disciplinary approach. Our paper is one valuable contribution in this direction. Many other efforts are required to approach a complete picture.

2) To which extent can such a study based on a piezo-electric material distinguish the two, given that any excitation concerns both?

In our piezoelectric system, the coexistence and strong coupling between elastic and electric fields makes it difficult to distinguish which field is driving the TLS.

We note that operationally, in applications of piezoelectrically coupled mechanical systems for quantum technologies it may not be relevant to distinguish between "acoustic" and "electric" interactions: TLS in the material would couple through either an electric or elastic interaction to the piezoelectric resonance, causing decoherence in either case.

Related question: which fraction of the energy of the oscillator energy is actually "mechanical"?

For a piezoelectric material, there is an electromechanical coupling coefficient (k_{eff}^2) which is the ratio of stored energy in the electric and elastic fields. This value depends on both piezoelectric constants of the material as well as design choices in the resonator and electrode geometry. For our particular device, k_{eff}^2 is approximately 25%.

3) The authors say in the abstract that previous studies were in the classical domain but there are numerous studies on qubits probing TLS defects either directly, indirectly, i.e. affecting mostly the resonator loss, and in both cases strongly and weakly coupled. In fact, superconducting qubits have become a standard tool to probe and control TLS - in some cases with fairly long coherence. So I think this needs to be put better in context or made

more precise. The introduction and reference list is focusses strongly on piezo-electric systems and would also benefit from including some of those studies - both early and recent.

We thank the reviewer for their suggestion. We have primarily focused our discussion of TLS literature on studies of mechanical systems. However, we agree with the referee that it is beneficial to include a wider review of TLS characterization techniques. We have added a number of citations on studies using superconducting qubits to the introductory paragraphs.

4) Given the importance of the results for the future of bosonic qubit encodings in microscopic mechanical oscillators I am surprised that there is no thorough discussion or outlook.

- What do the measured mechanical energy relaxation and dephasing times tell us?

The measured energy relaxation time $T1 \approx 10 \mu\text{s}$ and dephasing time $T2 \approx 1-10 \mu\text{s}$ for phonon states with mean occupation <10 are promising but leave room for improvement to enable bosonic encoding schemes. Please find further discussions below.

- Are the results better or worse than the ones in the literature? Can one compare to the equivalent $\tan(\delta)$ of relevant and well studied materials (piezo or not)?

The dephasing and decay times we measure are neither better nor worse – they are in line with literature, though we note that previous work had only measured these rates for qubits encoded in the 0 and 1 phonon state manifold (or much higher classical occupation). Our work shows that slower decay and dephasing times are possible when the information is encoded in coherent states containing a few phonons. We have explained why this improvement occurs in the manuscript through numerical modeling and an intuitive physical picture.

Yes, $\tan(\delta)$ measurements on LN made by Wollack et al (2021) can be used to compare to our results. In particular, our model uses a TLS density and coupling rate which is consistent with $\tan(\delta)$ measurements. Comparing across different materials is also possible, though outside the scope of our study.

- Why are they better or worse?

Thin film lithium niobate provides a highest ratio of piezoelectric coupling to loss that we are aware of in materials used in the field of NEMs and quantum acoustics (e.g. GaAs, AlN, Lithium Tantalate, GaN).

- Where are the TLS and are the numbers only relevant for a certain geometry or material?
As we describe in the text, some parameters are specific to our system, such as the simulated effective mass of the resonator, certain material properties of lithium niobate, and the loss tangent which is from measurements of lithium niobate oscillators of a similar geometry in Wollack et al (2021). The assumptions involved in the calculation are clearly delineated, allowing for readers to see which aspects are extensible to other device platforms.

- What are these timescales limited by and how can they be improved?

The purpose of the paper is to show that the nature of the dissipation – double-exponential, with an initial fast decay – is consistent with emission into TLS. We have added a line in the discussion (line 230) to highlight the need for improvements in fabrication and materials processing techniques.

- How much do they need to be improved to do something relevant (e.g. those applications mentioned in the conclusion)?

An in-depth study of applications requires systems-level and architectural considerations that go well beyond the scope of this work. Perhaps the most relevant answer we can provide the referee is from the work by Chamberland et al (2022). The horizontal line in the figure below is given by a coupling rate which we believe is possible with the piezoelectric coupling achievable here. The circle, square, and star show intrinsic $Q_s \sim 10^5$ - 10^8 are needed to realize the three regimes of interest in this paper – requirements for achieving fault tolerance in the architecture studied there. Our results are about an order of magnitude away from the lower end shown here.

Chamberland, Christopher, et al. "Building a fault-tolerant quantum computer using concatenated cat codes." PRX Quantum 3.1 (2022): 010329. Note that the above paper assumed that there would not be any excess low temperature dephasing – consistent with the standard TLS model. Our work points to the presence of these dephasing processes which may motivate new encodings and processes.

- How realistic is this?

We believe that improving the surface and material qualities in these resonators can help us mitigate the effects of TLS. As with all high-risk research endeavors, there is risk involved and we are not sure how to precisely quantify realism. Recent progress in the lab has been promising.

5) The authors mention 2 studies that observed non-exponential energy relaxation of qubits due to other reasons but there are also studies showing this due to TLS. A few recent examples from exponential to bi-exponential to oscillatory (depending on coupling strength to the strongest coupled TLS) are <https://arxiv.org/abs/2204.00499>, <https://arxiv.org/abs/2206.14104>, <https://www.nature.com/articles/ncomms7182>, <https://www.nature.com/articles/ncomms7182>.

Thank you! We have added a reference to the first two (arXiv) papers in our discussion of non-exponential qubit decay.

We note that the last two links (Nat. Comms.) refer to the same paper, and there is no mention of double-exponential decay in this paper. In fact, this paper reports that they observe *purely exponential* decay in the qubit when it is off-resonant with a TLS, and Rabi oscillations when it is on resonance (Fig. 2 caption). Our paper has addressed the possibility of Rabi oscillations on line 120, we do not observe such oscillations in our system.

With regards to that, can the authors clarify why Eq. 2 (biexponential) is more appropriate compared to a double exponential used in Ref. 12 to describes a primary fast dissipation channel (e.g. the strongest coupled TLS) together with a slower one (e.g. the ensemble formed by the other weaker coupled TLS together with other residual loss)? Is it related to the fact that here they study the relaxation of a harmonic oscillator? Or would both be largely equivalent and equally compatible to the numerical model? Related to that, what is the physical interpretation of the two coefficients a_1 and a_2 in Eq. 2?

We appreciate the reviewer raising this important point. Our use of a biexponential decay model is motivated by the proposed physical mechanism of phonon emission into an

ensemble of saturable TLS defects. We summarize the argument for why biexponential decay matches the intuitively expected dynamics:

- Fast initial decay of the mechanical state due to emission into the TLS
- Slower decay once the TLS become saturated and dephased and can no longer accept more phonons
- This conclusion is supported by the absence of any observed coherent oscillations between the mechanics and TLS.

In contrast, a double exponential decay, $P(t) = \exp(-\lambda (\exp(-t/T_1) - 1) \exp(-t/T_2))$, which was introduced in by Pop et al 2014, is justified by the dynamics of quasi-particle tunneling across a junction. This physics is unrelated to TLS as far as we could understand. The double exponential model was applied to TLS in the paper by Zemlicka et al -- in that paper states they are applying that formula "phenomenologically" to their TLS problem. This may be because at the time of the writing of their paper, this double exponential model was the only model that had been used in the field.

In our view, the biexponential form which we have introduced provides a more suitable empirical model for our data that is consistent with the conceptual framework of phonon emission into saturable TLS.

The coefficients a_1 and a_2 are related to the number of excitations exchanged in the system. The boundary condition at $t=0$ requires $a_1 + a_2 = n_0$, the initial phonon state size. In a limit where the initial phonon state is larger than the number of TLS ($n_0 > N$), then saturation occurs when the TLS have removed on average $N/2$ excitations from the mechanical mode, after which the fast decay transitions into slow decay.

Thus, the relation between the weights (a_1, a_2) and the initial state size (n_0) contains information about the number of TLS. In the regime of our experiment, the initial state size and TLS number appear to be in roughly the same range. Extrapolating the trend in the weights (Fig. 4c) suggests that further increasing the initial state would lead to growth in a_2 (the coefficient of slow decay) with n_0 , and a constant a_1 (coefficient of fast decay, which would be proportional to the number of TLS). In the limit of very large $n_0 \sim 100-1000$ phonons $\gg N_{\text{TLS}}$, we would see a transition into single-exponential decay.

In Fig. 4b it looks like that an actual double exponential $n_0 \exp[a (\exp(-\kappa_1 t) - 1)] \exp(-\kappa_2 t)$ (as derived in Ref. 12 for quasiparticle induced decay) rather than a biexponential as in Eq. 2 (maybe consistent naming would help) might fit better with fewer parameters as the fast decay of the numerics appears to be consistently faster (a log scale

would be beneficial to better see this). One would obtain different fitted time constants however - would the interesting size dependence observation still hold?

As stated above, we disagree with the reviewer on this point. First of all, contrary to the reviewer's claim, both models require the same number of parameters (3 parameters or 4 depending on how we count n_0 which forces the value of a_1+a_2). But more importantly, the double exponential model is not physically motivated by the problem at hand and therefore doesn't capture some effects.

For this discussion, we will again use the reviewer's nomenclature of double exponential referring to the expression derived for quasi-particle tunneling, and biexponential decay referring to our proposed model.

We plot the double exponential model (with 3 params) for $n_0 = 1, 2, 3, 50$

$$n_{DE}(t, n_0) = n_0 \times e^{(a \times (e^{-\kappa_1 t} - 1))} \times e^{-\kappa_2 t}$$

In contrast, the biexponential model (also with 3 params) for $n_0 = 1, 2, 3, 50$ give us:

$$n_{BE}(t, n_0) = a \times \exp(-\kappa_1 t) + (n_0 - a) \times \exp(-\kappa_2 t)$$

A key feature of the biexponential model is that the initial fast decay is limited by the number of TLS – the initial drop in $n(t)$ can go so far as to populate the TLS, and so the magnitude of the drop should asymptote to the number of TLS, as the initial excitation n_0 is increased. This is as one would expect from the physics of the process, and allows the eventual transition into a single exponential, as described above. At high drive powers, the TLS model and experiments agree on observing a single decay time – this is reproduced by the biexponential model. **The double exponential does not capture this feature, with the initial fast drop scaling with the excitation number n_0 instead of the number of TLS.**

6) It is unclear to me how the randomness and the known long term time variation of the TLS properties play a role in both the experiments and numerical simulations.

- Are these averaged measurements? If yes, how many averages and over what time scales are they obtained? Do the authors see differences when the experiment is repeated? The measurements are averaged. Each individual ringdown measurement of Fig. 2d represents data taken over a window of ~8-12 hours, with each data point averaged on the order of 10^4 times. Additionally, data is taken over a spread of time spanning multiple months. The double exponential behavior is repeatable with time constants and weights roughly consistent between experimental trials, even separated by weeks or months.

- Are the results an average of multiple initializations? Are the TLS parameters drawn randomly from a defined probability distribution of realistic detuning, couplings and decoherence times? If yes which distribution? If not, how robust is the observed double exponential behavior?

We assume the “results” in this question refer to the simulation results, since the experimental results were addressed in the prior bullet point. Our simulation is a Lindblad master equation solver in the QuTiP package (referenced in the paper), which returns the expectation values of the quantities of interest. The simulated results we plot in Fig. 4 represent two separate instances of the simulation, with different parameter sets. One parameter set (Fig 4b,c) with weaker coupling, and another (Fig 4d-f) with stronger coupling. This is specified in the figure caption, and in the main text on lines 181-202.

As we explain on line 166, in each execution of the simulation, the TLS are taken to be identical in their detunings, couplings and decoherence times. This is meant to simplify the model and to capture the average behavior of the TLS ensemble. Our rationale for choosing the values of each simulation parameter is explained in detail in lines 181-190 of the main text as well as section IV of the supplementary information.

The referee points out the benefit of consulting a probability distribution for realistic parameter values. In supplementary section IV we have explained how we approached this

process, and in supplementary figure S3 we plot the distributions that informed our choice of parameters.

How robust is the dephasing increase with n_0 (or under which assumptions/conditions is this predicted)?

In Fig. 4e, we plot two simulated results: one with the weakly coupled parameters (blue line) and one with the strongly coupled parameters (green line). In both of these regimes, the simulation predicts a reduction in the dephasing rate with larger n_0 .

We note that the behavior we observe is also consistent with a generalized tunneling model proposed by Faoro et al (2015) which predicts reduction of frequency noise for larger input powers, and has seen agreement with experimental results in superconducting systems.

7) One important new aspect is the size dependence of the observed dissipation including the not so intuitive finding that larger states both dephase and decay slower. Even though this is mentioned a few times and even though in the text a size dependent analysis is promised, I do not see clear data, figures, analysis and discussion of this aspect when it is mentioned. Later in Fig. 4c one can see that both weights increase - what does this mean? The primary decay tends to decrease with large error bars (is that meant?) and the secondary decay actually increases with size.

This size dependence refers to the trend in the weights a_1 , a_2 and rates κ_1 , κ_2 as well as the mechanical dephasing rate γ_{2m} over a range of initial state sizes. Figure 4c provides a direct comparison of our experimentally observed values with the simulated values, showing agreement over a range of initial state sizes. As mentioned above, the trend in the weights a_1 and a_2 , i.e. the saturation of a_1 and continued rise of a_2 , suggests that for a sufficiently large initial state size, the physics would eventually be described by single-exponential decay with the slower decay time as we would intuitively expect.

The same for the dephasing, I fail to see the data for the discussion in line 155 ff (I presume Fig.4e is referred to). Is it statistically significant compared to variations, e.g. for various repetitions or averaging times of the experiment? or between devices?

We have added a reference to Fig. 4e to this particular discussion.

The dephasing measurements are carried out over a time frame of hours to days. The interferograms are interleaved in time (e.g. not conducted in a linear chronological order) so

significant drifts of the TLS ensemble would appear as discontinuities in the data, rather than as a false correlation with initial state size. Given the amount of time required to complete these measurements, the dephasing effects we observe capture a time-averaged effect of the TLS on the mechanical mode.

We note that the plot in Fig. 3e includes two data points at $\tau = 0$. These two interferograms were taken 9 days apart in time, and the two points share significant overlap in their statistical uncertainty.

We do not perform dephasing measurements on multiple mechanical devices. However, we perform a few ringdown measurements on the lower-frequency mechanical mode (resonant frequency 2.05 GHz), and we observe double-exponential behavior for this device as well, with initial fast decay eventually giving way to slower decay.

Specific minor comments that should be addressed:

- abstract: is->are

We have corrected the grammatical choice.

- abstract: I'd mention piezo-electrical coupling

We have added the specification of piezoelectric coupling.

- references are not in order

Thank you, this artifact is not present in our arXiv preprint and will be fixed in the next iteration of the journal submission.

- introduction: please define the emphasized "quantum nonlinear dissipation" term and how this is applicable here and different to before

This term refers to the decay of the n^{th} level of a harmonic oscillator. This is typically given by $\kappa_n = n\kappa$: the dissipation of the quantum energy levels is described by a linear dependence on n . In our case, the decay process is more complicated, because the mechanical dissipation rate changes as the size of the phonon state changes, and so this simple linear description does not hold. We have added further explanation to the end of the introductory paragraphs: "the mechanical dissipation rate changes with the evolving phonon state size, and cannot be described by the simple linear relation expected for a harmonic oscillator."

- the device description contains quite some fabrication details - are they new and needed at this point?

We include further detail on fabrication in this paper, compared to the prior paper from this device (Ref. 43, Wollack et al 2022), because the focus of the paper is TLS, whose presence

is strongly tied to fabrication processes and material choices. Certain details are new (annealing and MgO-doped material), and we include all process details because they are relevant to the paper's particular subject of study.

- Fig. 2, maybe clarify early on that \bar{n} is the mean photon number and consider explicitly indicating n_0 in the plot (or have a legend).

The symbol \bar{n} is defined as the mean phonon number in the abstract, in the first introductory paragraph, and also in the first paragraph discussing the ringdown measurement (line 105). We have added a definition in the figure caption for both \bar{n} and \bar{n}_0 . We have also made the definition of \bar{n}_0 on line 115 more explicit.

Also, how large is the deviation from the Poissonian distribution or is it imposed to be that in the fit? Is that deviation part of the error bars shown in the measured distribution?

The fit does not impose a Poissonian distribution, or any distribution on the relative $P(n)$ values. The error bars are not related to the distribution; they reflect the uncertainty of each fit parameter, reported as one standard deviation extracted from the covariance matrix. The deviation from the Poissonian distribution is inferred to be a result of the interaction between the mechanical mode and the TLS, and the deviation is shown in Fig. 3f as well as supplementary Fig. S2.

- line 90: what would be too much coupling? Also, I'd assume similar measurements and insights could be obtained in the resonant limit?

To operate in the dispersive limit requires $|\Delta| \gg g$, and there is a somewhat flexible interpretation as to the precise ratio (Δ/g) that satisfies this. The modes of the qubit and mechanics are hybridized to a degree that depends on this ratio. It is typical to define a mixing angle, $\theta = 0.5 \cdot \arctan(2g\sqrt{n}/\Delta)$ which describes the hybridization of the qubit with the n^{th} level of the harmonic oscillator. When the two systems are resonant, the dressed states are fully mixed, e.g. each mode represents 50% qubit and 50% mechanics.

Indeed, it is possible to obtain information about the mechanical state through a resonant interaction with the qubit, see e.g. the work of Satzinger et al (2018) or Chu et al (2018). However, that measurement is not dispersive, and thus is not quantum nondemolition with respect to the mechanical state, which is an advantage of our measurement approach.

- line 99: Is the constant κ choice needed to fit the data successfully? Or in other words, was the linear n dependent κ needed to fit earlier (smaller phonon number) Ramsey measurements to successfully fit the data?

The model with n -dependent κ which we used in an earlier publication (Wollack et al 2022) was chosen because it was shown to fit the data better, resulting in lower MSE for all data sets being studied. However, even that model is still an approximation of the physical

process, as it does not account for detailed balance. For the particular data sets of the prior study, the model remains a very good approximation and yielded fit results with lower MSE and better stability than the general kappa model.

For this paper, using the n-dependent kappa model resulted in unstable and non-physical fit results. This may be because the phonon state sizes in this paper are larger than in the 2022 paper, yielding a greater consequence of enforcing the linear n-dependent decay of the harmonic oscillator to approximate the mechanics interaction with the TLS bath. This is discussed in section II of our supplementary information.

- line 108: is there something we learn from the observed time evolution of the distribution? e.g. Poissonian vs. thermal?

This interpretation is correct: the distribution at short time delays (e.g. short interaction with the TLS) closely follows a Poissonian distribution, while at larger delays, the mechanical $P(n)$ diverges from the Poissonian form and more resembles a thermal distribution. This is shown in Fig. 3f and supplementary Fig. S2, and is discussed in the associated captions as well as lines 146-154 of the main text.

- line 112: saturation dynamics vs. size is promised but I miss to see data how the 4 fit parameters in Eq. 2 change vs. initial phonon occupancy as well as (or at least or) a qualitative interpretation of it. also, define "saturation dynamics".

We use "saturation dynamics" in a general sense referring to the mechanical mode's initial period of fast decay eventually giving way to slower relaxation after the TLS have been saturated. This can, to some extent, be quantified by the trend of the fit parameters a_1 , a_2 and κ_1 , κ_2 when plotted against the initial state size, which shows agreement between our experimentally observed results and the simulated results.

We have added to the text to clarify this point:

Moreover, the trend in the weights a_1 and a_2 suggests that for a sufficiently large initial state size, the contribution of the fast initial decay (a_1) saturates while the contribution of the slow secondary decay (a_2) continues to increase. This indicates that above a certain threshold, the physics would eventually be described by single-exponential decay with the slower decay rate κ_2 , as we would intuitively expect once the TLS defects become saturated. This behavior is qualitatively different from the double exponential decay model that has been used to capture effects due to quasiparticles~\cite{agnetta_please_cite_something}.

- line 121: The explanation given requires that the T1 time of the TLS are longer than the energy relaxation time of the mechanical oscillator - is this correct?

It requires that the TLS are long-lived, although not necessarily longer-lived than the slow relaxation in the mechanics. The resonant TLS are expected to be long-lived in our system because the phononic bandgap offers protection from the phonon decay channels that are thought to dominate TLS relaxation. (See e.g. the supplement of MacCabe et al (2020) for discussion of bandgap effects on TLS.)

- line 134: isn't it mostly energy relaxation that causes the shift of the circle (center) rather than dephasing? Also in Fig. 3biii.

Even in the absence of energy relaxation, the spreading of the circle due to dephasing can be detected by our interferometry protocol through reduced oscillation amplitude. See supplementary information, section III, "Coherent state dephasing" for a derivation of this. We extract T_2 (as opposed to T_ϕ) from this protocol, so the effect of T_1 also comes into play.

- Is Fig. 3c plotted assuming $1/T_1=0$?

Yes, this cartoon is meant to illustrate the effect of pure dephasing, e.g. infinite T_1 . We have updated the caption to make this more clear.

- Fig.3e inset. Does the result qualitatively agree with the expectation of T_1 ?

Yes – comparison of Fig. 3e inset with the red data set in Fig. 2d (which has a similar initial state size) shows consistent behavior.

- line 161: does that mean the TLS bath would be locked to the coherent drive?

We believe so.

- line 177,178: "Similar to the experiment, the model shows an ..."

We have made this change.

Reviewer #3:

In this work the author uses a hybrid micromechanical- SC qubit qubit system to study phonon coherence and the effect of TLS. The qubit is only used as sensor (for Ramsey spectroscopy of the mechanical resonator) so -unless I am missing something- it is not actually very important as such; this is really a paper about TLS in mechanical systems which is a field that in some ways goes back to the 70s with some very impressive experiment having been done in recent years.

This is a clever and in many ways impressive experiment. Getting all the pieces of such a complicated experiment to work together and get reasonable data is far from trivial.

Focus in this work has been on looking on how the energy decay and dephasing depends on the initial phonon state size. While this is an interesting problem the results are -as the authors also state- somewhat inconclusive. The simple model used by the authors does agree with some of the data, but there are only deviations, and I am not entirely convinced by some of the parameters they extract (please see below). This means that while the data looks very nice, I am not entirely sure what we re meant to learn about TLS physics from the paper. That said, the measurement technique is very interesting.

A few specific comments:

-Double exponential decay is frequently seen in SC qubits, and it is well-known this can be due not only to quasiparticles (which is stated in the manuscripts) but also TLS. Hence, this in itself is not new (although this is a mechanical system, so there is a difference)

We have added a note in the paper regarding qubit double-exponential decay into TLS, with the appropriate references. We do emphasize that this behavior for a mechanical system, especially operated in a limit of small phonon number, is novel.

-The authors do not really discuss the effects of temperature in the paper. With a fridge temperature of 10 mK we would expect the sample temperature to be somewhat higher (say 50 mK). Now, this corresponds to an energy of about 1 GHz. We would therefore expect all "simple" two-level systems with Δ much lower than this to be saturated (of course we can still have "classical" fluctuators, but they would not behave as simple TLS).

Now, using their model the authors extract values for their 5 strongly coupled TLS of about 100 kHz ($\ll 1$ GHz). I am quite possibly missing something here, but I don't understand how this works. Again, even at 10 mK we know there are fluctuating TLS (sometimes known as fluctuators), but these are "classical" and not modeled in the way it is done in this paper (they are usually seen because they interact with TLS with much higher energies that are coherent at 10 mK)

Again, maybe I am misunderstanding something, but regardless I don't think this is clearly explained in the paper.

We apologize for the confusion, the parameter $\Delta_{\text{TLS}}/2\pi = 100$ kHz that we use for our simulations is the detuning of a resonant TLS from the mechanical system. It is not the TLS asymmetry energy, which is related to the TLS eigenenergy (in our paper, to denote the asymmetry energy, we use the symbol Δ_{as}). We believe that this has led the reviewer to believe that we are considering TLS with transition energies $\ll 1$ GHz and trying to

understand how they affect dynamics of the resonator. We are in fact only considering nearly resonant TLS. We have attempted to clarify this a bit in the text.

To summarize, we use this Delta (specifically, Δ_{TLS}) in the manner typical for Jaynes-Cummings physics and circuit or cavity QED: it denotes the *detuning* between the TLS and the mechanical eigenmode. This is defined in the paper on lines 167-168. Therefore $\Delta_{\text{TLS}}/2\pi = 100$ kHz refers to TLS with transition energies within 100 kHz of the mechanical resonance at 2.34 GHz. These TLS are not thermally saturated at the ~ 10 mK temperature of the dilution refrigerator.

REVIEWERS' COMMENTS

Reviewer #2 (Remarks to the Author):

I thank the authors for the detailed answer. All questions have been adequately addressed. The paper is ready for publication.

Reviewer #3 (Remarks to the Author):

The authors have revised the manuscript and answered the main question in my original review.

As previously mentioned, this is impressive and interesting work and I therefore now recommend that it is published.